# A GWAS Suggesting Genetic Modifiers to Increase the Risk of Colorectal Cancer from Antibiotic Use

**DOI:** 10.3390/cancers17010012

**Published:** 2024-12-24

**Authors:** Litika Vermani, Alicja Wolk, Annika Lindblom

**Affiliations:** 1Department of Molecular Medicine and Surgery, Karolinska Institutet, 17176 Stockholm, Sweden; 2Unit of Cardiovascular and Nutritional Epidemiology, Institute of Environmental Medicine, 17177 Stockholm, Sweden; alicja.wolk@ki.se; 3Department of Clinical Genetics and Genomics, Karolinska University Hospital, 17176 Stockholm, Sweden

**Keywords:** GWAS, colorectal cancer, antibiotics, genetics, loci, genes

## Abstract

We conducted a genome-wide association study to investigate the increased risk of the development of colorectal cancer associated with the frequent use of antibiotics. This case-control study of colorectal cancer cases and healthy controls involved the search for potential chromosomal regions associated with an increased risk of colorectal cancer due to frequent antibiotic use. To further validate the findings, a case-case study was undertaken comprising patients with frequent antibiotic use as cases versus patients with limited exposure to antibiotics as controls.

## 1. Introduction

Colorectal cancer (CRC) is a multifactorial disease caused by genetic and lifestyle risk factors [1]. Three major pathways resulting in the development of CRC include the adenoma–carcinoma pathway, the serrated pathway and an inflammatory pathway [1]. CRC is a complex and molecularly heterogenous disease. A subset of CRC is caused by predisposing genetic factors such as germline mutations in the DNA mismatch repair genes and the *APC* gene [2]. Approximately 25% of CRC cases occur in individuals with a family history of the disease but with unknown genetic syndromic status [3]. The remaining 60–65% of CRC cases arise sporadically due to acquired somatic genetic and epigenetic changes, which are largely attributable to potentially lifestyle-associated risk factors [3]. Lifestyle risk factors such as an unhealthy diet, obesity, lack of physical activity, smoking, alcohol consumption [3] and antibiotic use have been identified as potential contributors to CRC risk [4].

Worldwide overuse of antibiotics is a major public health challenge. Over the years the association between the onset of CRC and antibiotic use has been assessed by various studies and the results support the positive correlation between the use of antibiotics and the onset of CRC [5,6,7]. A Swedish nested case-control study found that antibiotic use was linked to a higher risk of developing colorectal polyps, suggesting the role of intestinal dysbiosis in the early stages of colorectal carcinogenesis [6]. The use of antibiotics such as quinolones and sulfonamides has been associated with an increased risk of proximal colon cancer among females. On the other hand, a reduced risk of rectal cancer has been observed among individuals with frequent use of antibiotics such as nitrofurantoin, macrolides and/or lacosamide, and metronidazole and/or tinidazole [4].

The available evidence suggests that the use of antibiotics could induce microbiome dysbiosis (an alteration in the microbiome of the gut), one of the important drivers of the initiation and progression of CRC [8,9]. The gut microbiota interacts with the colonic epithelia and immune cells of the host by releasing several metabolites, proteins and macromolecules which could have a potential role in the onset and regulation of CRC development [10].

The current study was conducted to search for potential genetic markers that could indicate an increased risk of CRC in individuals exposed to frequent antibiotic use. Specifically, the aim of the present study was to, for the first time, identify specific chromosomal regions harboring genetic elements that might interact with antibiotic exposure and potentially increase the likelihood of the development of CRC. The study aims to provide valuable insights into the risk of CRC development by determining whether the frequent use of antibiotics could lead to a genetic predisposition towards the disease. The insights of this study could help to enhance our understanding of the interplay between the genetic susceptibility of the development of CRC and environmental exposures such as the use of antibiotics.

## 2. Materials and Methods

### 2.1. Cases and Controls for First GWAS Analysis

#### 2.1.1. Cases

The cases included in our study were from the Swedish Colorectal Cancer Low-risk study, which comprised more than 3300 consecutive CRC cases recruited from 2004 to 2009, as previously described [11]. The study participants were interviewed by the same interviewer to gather information about their family history of CRC and other malignancies. Pedigrees were constructed based on this information for the families of the study participants. The study collected information on cases of CRC from first-degree relatives (parents, siblings and children) and second-degree relatives (aunts, uncles and grandparents) of the index persons, as well as cousins. Confirmation of CRC cases among relatives was done through medical records or death certificates. Participants without relatives with CRC were categorized as sporadic cases [12]. Participants recruited in 2004–2006 filled out a self-administered lifestyle questionnaire (*n* = 1767). The response rate for the questionnaire was 93% (*n* = 1639). The question asked in the questionnaire regarding antibiotic use included how often the participant was exposed to antibiotic use with options such as “yearly”, “sometimes” and “rarely” (coded as high, low, and no exposure). Out of 1639 CRC patients, 240 cases with Dukes A-D CRC used antibiotics frequently. The patients were genotyped at the Center for Inherited Disease Research at Johns Hopkins University, U.S.A. using the Illumina Infinium^®^ OncoArray-500K [13,14]. Genotype data was available for 143 patients with frequent antibiotic use and hence could be used in the study.

#### 2.1.2. Controls

The controls were genotyped in the same center in the U.S.A. as the cases using the same SNP chip [13]. The controls for this study were the same 1642 controls previously used for haplotype GWAS of all cases and controls [12]. Those were 536 cancer-free spouses to patients and 1106 blood donors. The controls only gave blood, and no questionnaire was filled in [12]. The controls were of Swedish origin.

### 2.2. Cases and Controls for Replication GWAS

#### 2.2.1. Cases

In the second haplotype GWAS analysis, the CRC-affected subjects were the same as those used in the first analysis (143 cases with frequent antibiotic use).

#### 2.2.2. Controls

Out of the 1639 CRC patients who provided the self-administered lifestyle questionnaire data, 532 cases with Dukes A-D CRC consumed limited antibiotics. Out of 532 CRC cases, genotype data was available for 472 patients with limited exposure to antibiotics and hence could be used in the replication study.

### 2.3. Statistical Methods

Two haplotype GWAS were performed. the first one to examine the association between the frequent use of antibiotics and a genetic susceptibility to CRC development and the second one to replicate the first analysis. The replication used as cases individuals with high exposure and as controls those with low exposure in a GWAS on the six chromosomes with statistically significant results in the first GWAS. The CRC case group comprised patients with stage I–IV disease and antibiotic use. The Uppsala Multidisciplinary Center for Advanced Computational Science (UPPMAX), which has high-performance computers for big-data analysis, was used to run haplotype GWAS. A sliding window method, with a window size ranging from 1–25 SNPs was used for the haplotype GWAS. The method tests all possible haplotypes from the first SNP to the 25th SNP, this generates various haplotypes of varying lengths for the same locus. A logistic regression method was employed to estimate the association between the chromosomal regions and CRC-affected subjects with the use of antibiotics and healthy controls. Both the haplotype GWAS analyses were performed using PLINK v1.07 software [15]. The analysis included the following parameters: hap-logistic (imputation by haplotype phasing based on multi-marker predictors), hap-window 1–25 (fixed number of SNPs in the sliding window model), Minor Allele Frequency (MAF) 0.01 (variants are filtered with MAF below the provided threshold), maximum per-SNP missing GENO 0.1 (removes all variants with missing call rates exceeding the default value 0.1). A *p*-value of <5 × 10^−8^ was considered statistically significant [16]. Haplotypes with an odds ratio (OR) > 1 were considered candidate risk loci.

## 3. Results

The clinical and demographic features of the case-control study with frequent antibiotic users as cases and healthy control persons with unknown exposure to antibiotics are described (Table 1). The clinical and demographic features of the case-case study with the same frequent antibiotic users as cases and 472 users with limited exposure to antibiotics as controls are described (Table 2).

A haplotype GWAS with 143 CRC cases involving frequent antibiotic use and 1642 controls was undertaken. A sliding window approach examined fixed-size windows of SNPs across the genome. The window slid across 1 to 25 SNPs at a time to assess how genetic variants within that window were associated with a trait. Numerous overlapping haplotypes with varying lengths and significance were generated in the analysis (Appendix A). Overlapping windows were used to examine different parts of the genome and identified regions with significant associations between regions, antibiotic use and the risk of CRC. The analysis generated data over all 23 chromosomes (Appendix A). In this first haplotype GWAS, six haplotypes at six different loci reached a level of *p* < 5 × 10^−8^ (Table 3). The loci were 2p13.3, 3q13.33, 4q32.3, 5q31.3, 8q21.11 and 11q22.1 (Table 3, Appendix A). The ORs were between 2.48 and 8.01 (Table 3). Of the six significant loci, two of them contained genes. The locus on 2p13.3 had the *ANXA4* gene, and the locus on chromosome 3 (3q13.33) had three genes *GPR156*, *LRRC58* and *FSTL1* (Figure 1). No genes were located within the regions on chromosomes 4, 5, 8 and 11. However, a pseudogene, *RPA2P3*, was in the locus 11q22.1 on chromosome 11.

To validate our initial findings, a subsequent case-case study was conducted in the same way as described for the first analysis and with the same 143 frequent antibiotic users as cases. As controls, 472 individuals from the same study but with limited exposure to antibiotics were used in this analysis (Appendix A, Table 4). The odds ratios (ORs) for the loci at 2p13.3, 3q13.33, 4q32.3, 5q31.3, 8q21.11 and 11q22.1 were between 2.29 and 6.12 and thus supporting the increased risk of CRC associated with these loci in those with frequent antibiotic use (Table 4). The *p*-values ranged between 1 × 10^−3^ and 2.95 × 10^−6^ and were less significant in this smaller experiment.

## 4. Discussion

Most CRC cases are sporadic in nature, which is largely associated with various lifestyle factors, including diet, physical inactivity, smoking and alcohol consumption. Among these modifiable factors, the use of antibiotics has emerged as a significant risk factor. Higher levels of antibiotic exposure have been linked to an increased risk of CRC development, as supported by multiple studies [4,17,18]. A study by Van der Meer J et al., 2021 was conducted to assess the relationship between oral antibiotics prescribed in Flemish general practice and the subsequent diagnosis of CRC, adjusting for comorbidities such as diabetes mellitus (DM). Their findings reported a 25% increased risk of CRC development over a 9-year period in people with antibiotic prescriptions compared to those with no prescriptions [19].

Our ability to fight infectious diseases that lead to morbidity and mortality has been revolutionized by antibiotics; but on the other hand, the widespread use of antibiotics has led to unintended consequences such as the disruption of microbial communities [20]. It is well-known that the long-term or recurrent use of antibiotics can lead to long-lasting changes in the gut microbiota that can potentially lead to increased cancer susceptibility [21]. Homeostasis of the gut microbiome can be disturbed by several factors such as age, lifestyle, diet, antibiotic therapies and physical activity that exerts an impact on the gut microbiome and equilibrium [21,22]. A symbiotic relationship exists between a human and their microbiome, wherein the human host provides a nutrient-rich habitat, and the microbiome performs various essential functions within the human body, which humans cannot achieve on their own.

The population of the gut microbiota is disordered in early life but eventually becomes stable over time, provided health is maintained and disruptions are minimal. Even with a few disturbances relating to exposure due to antibiotics, bacterial populations typically recolonize their niches, restoring their original composition and diversity; however, repeated disturbances can disrupt the microbiome leading to gut disorders and various other diseases [23]. The disruption of microbiota is known as dysbiosis. An overuse of antibiotics could modify the human microbiota composition; especially, gut microbiota dysbiosis could disrupt host immune homeostasis leading to an increased susceptibility to various diseases including cancer [24].

In the current study, we used haplotype analysis since this more detailed analysis has previously demonstrated better suitability for finding loci with higher odds ratios compared to SNP analysis [12,25,26]. Haplotype analysis could be done in homogenous populations. The cases were observed over a period from 2004 to 2006, and since the mean age of onset of CRC in Sweden is 73 years, most cases involved individuals born before 1950 and before the mass immigration to Sweden over the last few decades. Thus, the population at the time when the patients in this study were born was more homogenous than today. The only major ethnic immigrant group was Finnish at that time. We do not normally register ethnicity when recruiting patients in clinical studies in Sweden, but all participants were interviewed about family history and a rough estimation from the interviews and from their names indicated that less than 10% of the subjects were of Finnish origin and the rest were of Swedish origin. Breast cancer has an earlier age of onset than CRC, about two decades earlier, and Swedish breast cancer patients recruited around the same time have successfully been used for haplotype GWAS [25,26]. The main results of the breast cancer study confirmed, in more detail, the major results from the international consortium BCAC (Breast Cancer Association Consortium), which further suggests that also Swedish subjects born decades after 1950 are homogenous enough to be used in haplotype GWAS [24,25].

Here, haplotype analysis allows for the identification of clusters of SNPs closely linked and inherited together, suggesting that the interplay of multiple variants within these regions may influence their susceptibility to cancer. The interpretation of our results revealed a modifying effect on cancer risk caused by a genetic variant (or possibly variants) in the six haplotype regions identified. All six loci had high odds ratios, ranging between 3.65 and 8.01, suggesting a quite strong effect at each risk locus. Two of the loci contained genes.

The locus 2p13.3 had one gene named *ANXA4* within its haplotype region. The locus on chromosome 3 (3q13.33) had three genes *GPR156*, *LRRC58* and *FSTL1*. No gene was located within the regions on chromosomes 4, 5, 8 and 11. A pseudogene, *RPA2P3*, was located on chromosome 11 (11q22.1). Our results support the broader paradigm of cancer as a complex disease, which is driven by a combination of genetic predispositions and external lifestyle risk factors, emphasizing the importance of considering multiple genetic loci along with the environmental interplay in cancer research.

The replication cohort with the case-case analysis resulted in the anticipated outcome with mostly somewhat lower odds ratios. This was most likely because cases with less exposure to antibiotics could be associated with a less increased risk, which could explain why the difference in risk between high-exposure cases and low-exposure cases would be less significant. That the *p*-values were not at the level generally considered statistically significant was expected because of the smaller number of controls involved in the second analysis. In total, the results from the two association studies supported the hypothesis that there is an increased risk of development of CRC with frequent antibiotic use.

Both genes in the respective loci had already been suggested to be implicated in cancer. Annexin A4 (*ANXA4*) is a protein that binds to Ca2+ and phospholipids, and it is part of the Annexin family, which plays a role in the development of various tumor types through NF-κB signaling. *ANXA4* is highly expressed in multiple tumors and translocases between the membrane and cytoplasm in CRC. Higher *ANXA4* expression has been associated with increased CRC tumorigenesis and epithelial-mesenchymal transition (EMT) [27]. Annexins have been suggested to play crucial roles in the proliferation, invasion, metastasis and drug resistance of cancer [28,29]. The over-expression of *ANXA5* was observed in various tumor types such as breast cancers, hepatocarcinoma, CRC, cervical cancer, gastric cancer, pancreatic adenocarcinoma, prostate cancer, bladder cancer and nasopharyngeal carcinoma and has been reported to be involved in drug resistance in these cancers [28]. *ANXA5* could serve as a potential predictive biomarker for tumor development, metastasis and invasion, and could hold a prognostic and therapeutic significance for cancer [28].

The leucine-rich repeat gene family (LRR) class is involved in the activation of immune cells and could play a significant role in apoptosis and autophagy in CRC [30]. In the diversity and complexity of the tumor microenvironment, N6-methyladenosine (m^6^A) methylation modification plays a significant role. Key genes identified as m^6^A-modified genes, including the *LRRC58* gene, have been associated with CRC recurrence [31]. G protein-coupled receptors (GPCRs) are crucial for a wide range of cellular and physiological functions and are associated with many common human diseases. G protein-mediated signaling is increasingly linked to various aspects of oncogenesis, angiogenesis, immune evasion, metastasis and cancer drug resistance [32]. *FSTL1* is an extracellular glycoprotein and has previously been reported to play both oncogenic and tumor-suppressive roles in various cancer types [33].

Our finding supports the role of frequent antibiotic use in increasing the risk of CRC. Haplotype genome-wide association analysis could play a significant role in finding chromosomal regions that could potentially be associated with an excessive use of antibiotics and a risk of CRC in homogenous populations. The major limitation of this study is the small sample size and lack of information on the use of antibiotics among the healthy controls. The results could elucidate the complex relationship between genetic predisposition and the use of antibiotics in the etiology of CRC. Ultimately, these findings could increase the early detection of CRC and contribute to the development of personalized preventive strategies for individuals with a higher risk of CRC development due to frequent exposure to antibiotics.

## 5. Conclusions

The study identified six chromosomal loci associated with an increased risk (odds ratios 2.4–8.0) of CRC in individuals exposed to frequent antibiotic use. Two of the six haplotype regions harbored genes that have previously been reported to be associated with CRC tumorigenesis and EMT and CRC recurrence. The findings from this exploratory study should be confirmed by other larger studies with more detailed information on antibiotic use and information on CRC sub-sites to clarify the CRC risk related to the use of antibiotics and how it is influenced by genetic modifiers. The results may enhance the development of a personalized approach towards individuals with a higher genetic susceptibility to CRC when exposed to antibiotics, thus aiding in early detection and reducing the incidence of the disease.

## Figures and Tables

**Figure 1 cancers-17-00012-f001:**
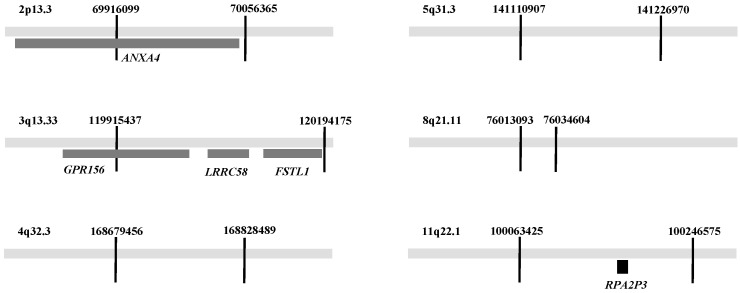
Figure showing statistically significant haplotype regions with base coordinates in GRCh 37 genome build.

**Table 1 cancers-17-00012-t001:** Demographic and clinical characteristics of cases with frequent antibiotic use.

	Demographics	Number
Gender	Male	75
	Female	68
Age range	<50 years	5
	>50 years	138
Family history	Sporadic	108
	Familial	35
Location	Left	86
	Right	25
	Caecum	15
	Unknown	17
Stage	Dukes A	18
	Dukes B	56
	Dukes C	47
	Dukes D	17
	Unknown	5

**Table 2 cancers-17-00012-t002:** Demographic and clinical characteristics of cases with limited use of antibiotics.

	Demographics	Number
Gender	Male	253
	Female	217
	Unknown	2
Age range	<50 years	25
	>50 years	445
	Unknown	2
Family history	Sporadic	368
	Familial	102
	Unknown	2
Location	Left	268
	Right	85
	Caecum	60
	Unknown	59
Stage	Dukes A	80
	Dukes B	176
	Dukes C	146
	Dukes D	50
	Unknown	20

**Table 3 cancers-17-00012-t003:** Chromosomal loci associated with CRC and frequent antibiotic use (first analysis).

Locus	SNP1-SNP2	F	OR	STAT	*p*-Value	Gene
2p13.3	rs12468494-rs2278933	0.04	3.65	32	1.5 × 10^−8^	*ANXA4*
3q13.33	rs2035669-rs4676787	0.01	8.01	33.5	7.1 × 10^−9^	*GPR156, LRRC58, FSTL1*
4q32.3	rs6834993-chr4_168828489_A_G	0.13	2.48	34.3	4.6 × 10^−9^	no gene
5q31.3	rs9686896-rs17208551	0.02	5.68	31.5	2 × 10^−8^	no gene
8q21.11	chr8_76013093_C_T-chr8_76034604_C_T	0.04	4.09	34.6	4 × 10^−9^	no gene
11q22.1	rs1452575-rs12225356	0.03	4.86	33.5	7.3 × 10^−9^	*RPA2P3* (pseudogene)

SNP1 (single nucleotide polymorphism 1): the first SNP of the haplotype region. SNP2 (single nucleotide polymorphism 2): the last SNP of the haplotype region. F: sample frequency. OR: Odds ratio. STAT: test statistic (T from Wald test).

**Table 4 cancers-17-00012-t004:** Chromosomal loci associated with CRC and frequent antibiotic use (second analysis).

Locus	SNP1-SNP2	F	OR	STAT	*p*-Value	Gene
2p13.3	rs12468494-rs2278933	0.05	2.63	14.3	2 × 10^−4^	*ANXA4*
3q13.33	rs2035669-rs4676787	0.01	6.12	12.8	4 × 10^−4^	*GPR156, LRRC58, FSTL1*
4q32.3	rs6834993-chr4_168828489_A_G	0.15	2.29	21.9	2.95 × 10^−6^	no gene
5q31.3	rs9686896-rs17208551	0.02	4.01	10.9	1 × 10^−3^	no gene
8q21.11	chr8_76013093_C_T chr8_76034604_C_T	0.04	4.38	23	1.63 × 10^−6^	no gene
11q22.1	rs1452575-rs12225356	0.04	2.68	10.5	1.21 × 10^−3^	*RPA2P3* (pseudogene)

SNP1 (single nucleotide polymorphism 1): the first SNP of the haplotype region. SNP2 (single nucleotide polymorphism 2): the last SNP of the haplotype region. F: sample frequency. OR: odds ratio. STAT: test statistic (T from Wald test).

## Data Availability

The datasets used and analyzed during the current study are available from the corresponding authors on reasonable request.

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
