# Peer review of "A GWAS Suggesting Genetic Modifiers to Increase the Risk of Colorectal Cancer from Antibiotic Use"

_cancers, 2024, doi:10.3390/cancers17010012_

Round 1

Reviewer 1 Report

Comments and Suggestions for Authors

The manuscript is written clearly and up to the point. I have only a few issues with the design of the study. I realize that recruiting such a high number of subjects for the study is an enormeous achievement -this goes primarily for the healthy controls, but information about their exposure to antibiotics would be useful for their preselection. Another issue is the spectra of the antibiotics used which affect different bacteria and create different dysbiosis profiles. It would be interesting if authors would provide information on which antibiotics were most frequent among their group of analyzed patients. I suspect that penicillin would be at the top of the list.

Minor issues- figure title should be under the figure and the figure 1 should be augmented for better legibility. 

Author Response

Thank you for critically reading the manuscript and suggestions how to improve it.

The manuscript is written clearly and up to the point. I have only a few issues with the design of the study. I realize that recruiting such a high number of subjects for the study is an enormeous
achievement -this goes primarily for the healthy controls, but information about their exposure to antibiotics would be useful for their preselection. Another issue is the spectra of the antibiotics used which affect different bacteria and create different dysbiosis profiles. It would be interesting if authors would provide information on which antibiotics were most frequent among their group of analyzed patients. I suspect that penicillin would be at the top of the list.

Thank you for asking about exposure, we agree that it is important information. The questions asked
in the questionnaire were – How often were you exposed to antibiotics, ¨Yearly¨, ¨Sometimes¨ or
¨Rarely ¨(coded as high-, low-, and no exposure).
We have added this information in Methods – lines 116-119 which now reads:

The question asked in the questionnaire regarding antibiotic use included how often the participant was exposed to antibiotic use with options as “yearly”, “sometimes” and “rarely” (coded as high-, low-, and no exposure).

We have no information on what kind of antibiotics was used.
No information at all from healthy controls.

Minor issues- figure title should be under the figure and the figure 1 should be augmented for better legibility.

Thank you for letting us know. We have now modified the figure for better resolution and quality and
also have moved the title under the figure. Please refer to figure 1 in the manuscript

Reviewer 2 Report

Comments and Suggestions for Authors

The authors obtained results on the association of certain SNPs localized in six chromosomal regions with an increased risk of developing colorectal cancer, and in 4 of these regions no gene was found or these regions contained pseudogenes. Moreover, for all six identified regions, the odds ratio varied between 3.6 and 8, which is a high value for this type of analysis and implies a strong association. In the second part of the work, where the authors analyzed SNP clusters in colorectal cancer patients who frequently and rarely use antibiotics, significantly lower odds ratio values ​​​​and less significant P-levels were obtained.

Disadvantages of this work:

1) Clinical data do not allow us to understand what kind of ethnic group it was. Meanwhile, usually for GWAS analysis, ethnic differences introduce additional variation in the results obtained, and the ethnic homogeneity of the sample in the case of colorectal cancer is generally critical.

2) My opinion may be subjective, but the sample size for such an analysis of several hundred patients may lead to erroneous conclusions. The sample is very small.

3) Frequent and infrequent use of antibiotics in patients included in the study - how much, for what period? There is not a word about this in the description of the methods.

Conclusion: the article needs to be revised and reviewed again.

Author Response

Thank you for critically reading the manuscript and suggestions how to improve it.  

The authors obtained results on the association of certain SNPs localized in six chromosomal regions with an increased risk of developing colorectal cancer, and in 4 of these regions no gene was found or these regions contained pseudogenes. Moreover, for all six identified regions, the odds ratio varied between 3.6 and 8, which is a high value for this type of analysis and implies a strong association. In the second part of the work, where the authors analyzed SNP clusters in colorectal cancer patients who frequently and rarely use antibiotics, significantly lower odds ratio values and less significant P-levels were obtained.

Disadvantages of this work:
1) Clinical data do not allow us to understand what kind of ethnic group it was.

All patients were considered with a Swedish background, a minor part (<10%) had Finnish
origin (from names, we do not ask for ethnicity)

2) Meanwhile, usually for GWAS analysis, ethnic differences introduce additional variation in the results obtained, and the ethnic homogeneity of the sample in the case of colorectal cancer is generally critical.

We agree that homogenous population is critical especially when we are analyzing haplotypes
and, in our study, we have included cases and controls both with Swedish background and a
minor part (<10%) had Finnish origin (from names, we do not ask for ethnicity).

3) My opinion may be subjective, but the sample size for such an analysis of several hundred patients may lead to erroneous conclusions. The sample is very small.

Yes, we agree, this is a limitation, and we have mentioned this in the manuscript in the lines 331-332
which read as:
The major limitation of this study is the small sample size and lack of information of the use of
antibiotics among the healthy controls.

3) Frequent and infrequent use of antibiotics in patients included in the study - how much, for what period? There is not a word about this in the description of the methods.

Conclusion: the article needs to be revised and reviewed again.

Thank you for asking about exposure, we agree that it is important information. The questions asked
in the questionnaire were How often were you exposed to antibiotics, ¨Yearly¨, ¨Sometimes¨ or
¨Rarely ¨(coded as high-, low-, and no exposure).
We have added this information in Methods lines 116-119 which now reads:

The question asked in the questionnaire regarding antibiotic use included how often the participant was exposed to antibiotic use with options as “yearly”, “sometimes” and “rarely” (coded as high-, low-, and no exposure).

Round 2

Reviewer 2 Report

Comments and Suggestions for Authors

The article has been slightly revised, but overall they do not change the impression of the manuscript. The main and significant comments on the article are:

1) The main study group is ethnically heterogeneous. To support my opinion, I cite open data from Wikipedia: Despite the traditional predominance of Swedes in the population of Sweden, the modern population of Sweden is quite diverse in racial and ethnic terms due to a new wave of political and economic immigration from developing countries. The country's population is actually divided into two large groups: autochthonous and immigrant.

As of 2023, the proportion of people who were born in Sweden and to whom both parents were born in Sweden was 65%, with one foreign parent - 7.8%, with both foreign parents - 6.6%. The proportion of residents born abroad is 20.5%. Among the autochthonous peoples, the Swedes and the long-time inhabitants of the northern regions of Sweden stand out — representatives of the Finno-Ugric peoples, the Finns and the Sami. Ethnic Swedes make up the majority, about 7.5 million people. More than 50 thousand indigenous Finns live along the border with Finland, which was once part of the Swedish Kingdom, and in the central regions of the country there are over 450 thousand people of Finnish origin who immigrated to the country during the 20th century, as well as their descendants.

In the materials and methods section of the study, nothing is written about how the supposedly homogeneous study group was formed (in the responses to the reviewer it is written that 90% are Swedes, and 10% are Finns, based on the last name). Such a formation of the research group for this type of study and this localization of cancer is unacceptable.

2) The authors themselves admit that the small sample size for both GWAS analysis options is the main drawback of this study. Thus, these 2 shortcomings are irreparable in the context of this article, and the conclusions drawn are a logical consequence of these shortcomings

Conclusion: reject the article.
